# E3 Ubiquitin Ligase PUB23 in Kiwifruit Interacts with Trihelix Transcription Factor GT1 and Negatively Regulates Immune Responses against *Pseudomonas syringae* pv. *actinidiae*

**DOI:** 10.3390/ijms25031930

**Published:** 2024-02-05

**Authors:** Tao Wang, Gang Wang, Jiyu Zhang, Jiping Xuan

**Affiliations:** Jiangsu Key Laboratory for the Research and Utilization of Plant Resources, Institute of Botany, Jiangsu Province and Chinese Academy of Sciences, Nanjing 210014, China; immmorer@163.com (T.W.); wg20092011@163.com (G.W.); maxzhangjy@cnbg.net (J.Z.)

**Keywords:** U-box, protein interaction, bacterial canker, immunity

## Abstract

Kiwifruit bacterial canker caused by *Pseudomonas syringae* pv. *actinidiae* (Psa) is the most serious disease threatening kiwifruit production. Our previous study found genes encoding the U-box containing proteins were significantly regulated by Psa infection. Here, we report a U-box type E3 ubiquitin ligase PUB23 in kiwifruit which acts as a negative regulator of immune responses against Psa. PUB23 was found to physically interact with GT1, a trihelix transcription factor, in vitro and in vivo. The expression of *GT1* was up-regulated in *PUB23*-silenced plants, indicating that interacting with PUB23 may directly or indirectly suppress *GT1* expression. The silencing of *PUB23* led to enhanced immune responses of PAMP-triggered immunity (PTI), including a higher expression level of defense marker genes *PR1* and *RIN4*, and increased accumulation of hydrogen peroxide and superoxide anion. Our results reveal a negative role PUB23 plays in kiwifruit immune responses against Psa and may regulate gene expression by interacting with GT1.

## 1. Introduction

Kiwifruit bacterial canker is the most serious disease that threatens the development of the kiwifruit industry. The pathogen *Pseudomonas syringae* pv. *actinidiae* (Psa) was first reported in Shizuoka, Japan, in 1984 [1] and has been detected in the main kiwifruit planting countries, including China, New Zealand, and European countries [2,3,4]. Due to the fact that pathovars of *P*. *syringae* have been found to cause serious damage to main economic crops in the world, a number of studies have been conducted to explore the pathogenesis of *P*. *syringae*. Now it is known that plant immune system is composed of PAMP-triggered immunity (PTI) and effector-triggered immunity (ETI). PTI perceives PAMPs by plant pattern recognition receptors (PRR) and triggers defense responses including production of reactive oxygen species (ROS) [5]. Most plant pathogens can be fended off by PTI. To evade PTI, pathogens deploy various effectors into the host cell [6] which are detected by intracellular nucleotide-binding leucine-rich repeat (NLR) proteins and trigger ETI. PTI and ETI can combine to cause hypersensitive response (HR) at infection site, which involves programmed cell death (PCD) to restrict the spreading of pathogen. In *Arabidopsis*, bacterial PAMP flg22 is sensed by its receptor FLS2 and triggers immune responses [7]. To prevent immune responses, flg22 recruits U-box E3 ubiquitin ligases PUB12 and PUB13 to ubiquitinate and consequently degrade FLS2 by 26S proteasome [8].

Our previous research found that Psa led to the up-expression of ubiquitin ligase-coding gene *PUB23* in kiwifruit which belongs to the plant U-box type E3 ubiquitin ligase family [9]. U-box is a derived version of RING-finger domain that lacks the hallmark metal-chelating residues but is likely to function similarly to the RING-finger in mediating ubiquitin-conjugation of protein substrates [10,11]. Ubiquitination regulates diverse cellular processes, including floral transition, circadian rhythm, photomorphogenesis, and cell death [12,13]. E3 ubiquitin ligase is responsible for attaching ubiquitin to the substrate protein in the ubiquitination pathway, which mediates a post-translational modification of cellular proteins, commonly targeting them for destruction by the proteasome. The level and placement of ubiquitination have various impacts. Polyubiquitination often promotes rapid degradation, while monoubiquitination or multiubiquitination can instead alter protein activity and interaction with other proteins [14,15].

Plant U-box protein 13 (PUB13) is one of the well-studied examples in plant immune system. In *Arabidopsis*, silencing of *PUB13* increased resistance to biotrophic pathogens but reduced resistance to necrotrophic pathogens by inducing spontaneous cell death [16]. PUB13 is also involved in regulating the FLS2-mediated PTI [17]. PUB17 of tobacco was found to involve in cell death by interacting with BTB-BACK domain protein POB1 [18]. Another member of this family, PUB20/CMPG1, is required for HR-PCD mediated by Avr9/Cf-9, Pto/AvrPto interactions, and recognition of the PAMP INF1 [19].

In *Arabidopsis*, PUB23 and its homologs PUB22 and PUB24 were induced by PAMPs and pathogen infection [20]. They act as negative regulators of PTI in *Arabidopsis* immunity. In this study, we explored the role PUB23 played in immune responses of kiwifruit to Psa infection. PUB23 was demonstrated to interact with a trihelix transcription factor GT1, which belongs to the GT factor family. The silencing of *PUB23* led to the up-regulation of *GT1* and defense marker genes *PR1* and *RIN4*, and increased production of reactive oxygen species in both susceptible and resistant kiwifruit cultivars. Our results indicated that PUB23 interacts with GT1 and acts as a negative regulator in kiwifruit immunity against Psa.

## 2. Results

### 2.1. Screening of Proteins Interacting with PUB23

Our previous work found that Psa induced obvious change in expression of 25 genes encoding E3 ubiquitin ligases, 22 of which were up-regulated [9]. Among these genes, *PUB23*, which encodes the plant U-box containing protein 23, has been reported to participate in plant–pathogen interaction in several species. To identify substrates of PUB23 protein in kiwifruit, we constructed the Y2H prey library with Psa-inoculated ‘Hongyang’ and ‘Jinkui’ cultivars, and recombined the pGBKT7-PUB23 bait vector. After library screening, 272 positive clones were able to regularly grow on SD/-Trp/-Leu/-His medium. All of these positive clones were then sequenced, and the result was 59 successful sequences. To verify the interaction between these positive clones and PUB23, we plated these clones onto SD/-Trp/-Leu/-His/Ade+x-α-gal mediums. Of the 59 clones, 58 clones could regularly grow on the SD/-Trp/-Leu medium; meanwhile, only 16 ones were able to survive and became blue on SD/-Trp/-Leu/-His/Ade+x-α-gal medium (Figure 1). BLAST results reveled that except for clone WT-180, which was not similar to any sequences in the database, the remaining 15 clones encode proteins participating in metabolism, signal transduction, organ development and disease resistance (Table 1).

### 2.2. Sequence Analysis of the Potential Interacting Protein GT1

Among the 16 potential clones, the colony of clone WT-194 was the bluest, indicating the most possible interactions with PUB23. Its encoding protein is a trihelix transcription factor, which is the number of GT factor family, so we named it GT1. GT factors have a conserved DNA-binding domain which contains three helical structures (helix–loop–helix–loop–helix) [21]. GT factors were first discovered to specifically bind to a light-responsive element [22] to regulate the light responses of plants [23,24]. Members of GT factor family have been found to regulate multiple processes of plant growth and development, such as floral organ development [25,26]. They are also related to biotic and abiotic stresses [27,28]. Based on the full amino acid sequences of GT1 and GT factors in other plant species, we created a phylogenetic tree (Figure 2). Among the species in the tree, GT1 was mostly close to ASIL2 in *Solanum tuberosum*, which plays roles in embryo maturation [29] and was far from ASIL1 in *Arabidopsis thaliana*. GT1 showed high homology with trihelix transcription factor A0A2R6RR72.1.A with the three-helix structural domain.

### 2.3. PUB23 Interacts with GT1 In Vitro and In Vivo

To demonstrate the direct interaction between PUB23 and the Y2H screened potential protein GT1, an in vitro pull-down assay was performed. PUB23 was expressed by cell free expression system as fusion protein FLAG-PUB23, and GT1 was expressed as His-GT1. The fusion proteins FLAG-PUB23 and His-GT1 were co-incubated and precipitated with FLAG-tag magnetic beads or His-tag magnetic beads, respectively. The eluted proteins were then immunoblotted with anti-FLAG or anti-His antibodies. As shown in Figure 3, the fusion proteins FLAG-PUB23, His-GT1, and the control His-GFP were all successfully expressed. FLAG-PUB23 was pulled down by His-GT1, which was adsorbed by His-tag magnetic beads in IP:His treatment, while FLAG-PUB23 was not pulled down by His-GFP. His-GT1 was pulled down by FLAG-PUB23 which was absorbed by FLAG-tag magnetic beads in IP:FLAG treatment, while His-GFP was not pulled down by FLAG-PUB23. In conclusion, PUB23 interacts directly with GT1 in vitro.

To detect whether PUB23 interacts with GT1 in planta, we performed co-immunoprecipitation assay in *Nicotiana benthamiana*. Recombined vectors FLAG-PUB23 and GFP-GT1 were constructed and infiltrated in *N. benthamiana* using Agrobacterium-mediated transient assay. The recombined proteins FLAG-PUB23 and GFP-GT1 and the control protein GFP were all detected in *N. benthamiana* (Figure 3). A smaller protein which showed the same characteristic with GFP-GT1 was also detected. It was supposed to be a truncated product of GFP-GT1. The results showed that GFP did not interact with FLAG-PUB23, while GFP-GT1 interacted with FLAG-PUB23. So it suggested that PUB23 interacts with GT1 in vivo.

### 2.4. Silencing of PUB23 Down-Regulated Expression of GT1, PR1, and RIN4

As E3 ubiquitin ligases, PUB proteins usually regulate cellular processes through the ubiquitination pathway by a post-translational modification of proteins, targeting them for destruction or altering their activities. The interaction between PUB23 and GT1 indicates that PUB23 may function though directly interacting with GT1 and degrades it or alters its activity of regulating the expression of other genes. So we silenced *PUB23* in kiwifruit by virus-induced gene silencing (VIGS) system. To examine the influence of PUB23 on GT1 and immune responses of kiwifruit, we detected expression of defense marker genes *PR1*, *RIN4*, and *ICS1* which is involved in the biosynthesis of salicylic acid (SA). As shown in Figure 4, the silencing of *PUB23* in both ‘Hongyang’ and ‘Jinkui’ induced the up-regulation of *GT1*, especially in ‘Hongyang’. So we suppose that PUB23 negatively regulates expression of GT1 by directly interacting with it. All of the three defense marker genes examined were up-regulated in *PUB23*-silenced ‘Jinkui’ kiwifruit, of which *PR1* was the mostly induced. In canker-susceptible ‘Hongyang’ kiwifruit, the expression pattern of the resistance genes was similar to that in ‘Jinkui’ with the exception of *ICS1*, which was down-regulated by *PUB23* silencing. The different expression patterns of *ICS1* need further research and may be associated with the accumulation level of SA.

### 2.5. Silencing of PUB23 Increased ROS Accumulation

Along with the induction of pathogenesis-related (PR) genes, ROS burst is one of the first reactions of plants upon pathogen infection. ROS is also involved in HR-PCD at the infection sites to prevent the spread of pathogens. To explore the biochemical mechanisms of kiwifruit plants in defense against Psa, we compared the accumulation of hydrogen peroxide and superoxide anion (Figure 4C,D) of *PUB23*-silenced plants with those of the control. The contents of hydrogen peroxide of *PUB23*-silenced plants including both ‘Hongyang’ and ‘Jinkui’ were higher than those of the control. *PUB23* silencing induced increased contents of hydrogen peroxide in both ‘Hongyang’ and ‘Jinkui’. Hydrogen peroxide content in *PUB23*-silenced ‘Hongyang’ was higher than that of ‘Jinkui’. It was the same case in the control plants. Superoxide anion contents of *PUB23*-silenced ‘Hongyang’ and ‘Jinkui’ were also higher than those of the control. ‘Jinkui’ accumulated more superoxide anion than ‘Hongyang’, in *PUB23*-silenced and control plants, respectively. It was the opposite situation for hydrogen peroxide.

## 3. Discussion

Kiwifruit bacterial canker is caused by Psa, a pathovar of *P. syringae* which brings about severe damage to economic crops in the world. Little is known about the pathogenesis of Psa, while many studies have been conducted on other pathovars of *P. syringae*, especially the model species *P. syringae* pv. tomato DC3000. *P. syringae* is an evolutionarily bacterial species complex with a remarkably broad host range. To colonize and parasitize host plants, *P. syringae* suppresses plant immune system by the pathogenicity factors delivered by a functional hypersensitive response and pathogenicity (*hrp* pathogenicity island [PAI]) type III secretion system (T3SS) [30]. Type III effectors suppress immunity and promote pathogenesis in a variety of plants but trigger ETI [31], which detects injected effectors in other plants. Consequently, type III effectors determine the pathogenicity of individual *P. syringae* strains in large part and restrict their potential hosts such that they cannot recognize their effectors. In plants, PTI is the second line of defense to pathogens. Before this response, plants firstly recognize the conserved microbial features PAMPs/MAMPs [32,33] by plant pattern recognition receptors (PRRs) and initiate the basal defense response PTI, which mediates relatively weaker immune responses with a broad-spectrum defense against pathogens. These responses include the activation of mitogen-activated protein kinases (MAPKs), rapid production of reactive oxygen species (ROS), and the induction of pathogenesis-related (PR) genes [34,35,36].

Several PAMPs and their corresponding PRRs have been identified and studied [37,38,39], including the U-box E3 ubiquitin ligases. In our previous study on kiwifruit, Psa induced regulated expression of a certain number of genes encoding proteins with U-box domain. In eukaryotes, ubiquitination is a key post-translational protein modification. This process is completed by three classes of enzymes: the ubiquitin-activating enzymes (E1), the ubiquitin-conjugating enzymes (E2), and the ubiquitin ligases (E3). Of the three enzymes, E3 plays a critical role in substrate specificity. The polyubiquitination of a given substrate serves not only as a signal for degradation but also for targeting and re-profiling [40,41]. Even the addition of a single ubiquitin moiety plays an important role in determining the fate of the substrate [42]. Several examples of PUB proteins involved in plant response to abiotic stresses were reported. In *Arabidopsis*, abiotic stresses induced rapidly and coordinately expression of *PUB22* and *PUB23*. Both PUB22 and PUB23 physically interacted with RPN12a, a subunit of the 19S regulatory particle in the 26S proteasome. PUB22 and PUB23 negatively regulated drought tolerance by ubiquitinating RPN12a for degradation [43]. Apple PUB23 was found to reduce cold-stress tolerance by degrading the cold-stress regulatory protein MdICE1 [44]. PUB23 and its homologs PUB22 and PUB24 were also involved in regulation of PTI in *Arabidopsis* [20]. The homologous PUB triplet acts as negative regulators of PTI in response to several distinct PAMPs. The PUB triplet repressed immune responses including the oxidative burst and the MPK3 activity, and transcriptional activation of marker genes were increased and/or prolonged. In the response of kiwifruit to Psa infection, expression of *PUB23* was found to be observably up-regulated. Other examples of PUB proteins participating in regulating plant immunity against pathogens were frequently reported. PUB proteins are found to regulate different immune responses, including PTI, ETI, and HR-PCD, which can be remarkably induced by PUB proteins, or suppressed. In tobacco, *Cladosporium fulvum* infection transiently induced expression of the PUB protein ACRE276, which improved the defense response by enhancing PCD. Its functional homolog PUB17 played the similar role in *Arabidopsis* resistance against *P. syringae* pv. *tomato* [45].

To investigate the role PUB23 plays in kiwifruit immunity against to Psa, we screened potential substrates interacting with PUB23, and ultimately identified the interacting protein GT1. GT1 belongs to the trihelix transcription factor family, which is named after its conserved three-helix structural domain (helix–loop–helix–loop–helix). Trihelix transcription factors are widely found in a number of plant genomes and are involved in various biological processes, such as embryogenesis, formation of perianth, stomata, and seeds [46,47,48]. They are also found to participate in regulation of plant responses to stresses especially abiotic tresses [49]. Transgenic *A. thaliana* with overexpression of *GT-2A* and *GT-2B* displayed enhanced tolerance to drought stress [50]. The BnSIP1-1 protein significantly increased the germination rate of oilseed rape (*Brassica napus*) seeds under osmotic stress, salt stress, and abscisic acid (ABA) treatments by regulating ABA synthesis and signaling [51]. In *Arabidopsis*, two trihelix factors, VFP3 and VFP5, were found to interact with agrobacterium virulence protein VirF, which was an F-box protein [52]. Studies suggest that VirF recognizes and induces degradation by E3 ubiquitin ligase complex and the ubiquitin/proteasome system (UPS) of the plant protein VIP1 and its associated bacterial effector VirE2 [53,54]. Our study confirmed that PUB23 physically interacted with GT1 in vivo and in vitro and expression of *GT1* was enhanced in PUB23-silenced kiwifruit plants. So it is supposed that PUB23 negatively regulates the expression of GT1 by physical interaction. Trihelix factors are known as GT factors because their conserved structural domain binds to GT elements which are regulatory DNA sequences usually found in the promoter region of many different plant genes. An *Arabidopsis* trihelix factor ASR3 was reported to bind GT-like elements in the promoter of the early immune response gene *FLG22-INDUCED RECEPTOR-LIKE KINASE1* and negatively regulated a large portion of genes responding to flg22 [55]. ASR3 phosphorylation by MPK4 through a mitogen-activated protein kinase (MAPK) cascade enhanced its DNA binding activity to suppress gene expression in patter-triggered immune responses. *PUB23*-silenced kiwifruit plants also showed elevated expression of pathogenesis-related genes *PR1* and *RIN4*. *PR* genes have been frequently used as marker genes for systemic acquired resistance in many plant species. Our previous study found that Psa induced differentially expression of PR genes of six PR protein families. In *Arabidopsis*, PAMPs induced RING-type E3 ubiquitin ligases AtRDUF1 and AtRDUF2 positively regulate the expression of *PR1* and PTI [56]. RPM1-Interacting Protein 4 (RIN4) acts as a vital defense regulator [57] and is targeted and modified by several pathogen effectors, such as AvrRPM1, AvrRpt2, AvrPto, and AvrPtoB [58], to destabilize plant immunity. RIN4-mediated immunity is regulated by a trihelix factor GTL1 which positively regulates defense genes and inhibits factors that mediate growth and development [59].

The accumulation of reactive oxygen species (ROS) is one of the first reactions of plants upon pathogen infection [60]. This process is usually rapid and transient, and is involved in PCD, which limits the spread of pathogens at the infection sites. Our results showed that the accumulation of ROS was enhanced in *PUB23*-silenced kiwifruit plants, indicating a negative regulator of PUB23 in ROS production.

Our findings show that the U-box type E3 ubiquitin ligase PUB23 in kiwifruit, which is induced by Psa infection, interacts with a trihelix transcription factor GT1. The expression of *GT1* was increased in *PUB23*-silenced plants, indicating the interaction with PUB23 suppressed GT1 expression. The silencing of *PUB23* led to the higher expression of defense marker genes *PR1* and *RIN4*. The accumulation of ROS including hydrogen peroxide and superoxide anion was also enhanced. All these results indicate that PUB23 acts as a negative regulator of the kiwifruit immune response against Psa.

## 4. Materials and Methods

### 4.1. Plant Materials and Treatments

Two kiwifruit (*Actinidia chinensis*) cultivars were used in this study: canker-susceptible cultivar ‘Hongyang’ and canker-resistant cultivar ‘Jinkui’. Vanes of the two kiwifruit cultivars were three years old and trained on pergola system. All the vanes were kept in Institute of Botany, Jiangsu Province and Chinese Academy of Science, China. Shoots were collected from ‘Hongyang’ and ‘Jinkui’, respectively, placed in MS medium, and maintained in a growth chamber with a temperature of 25 °C and 12 h/12 h (light/ dark) cycles. After two days, the shoots were inoculated with *Pseudomonas syringae* pv. *actinidiae* (Psa). Stems of the shoots were carved with a knife and injected with water (control) or Psa which was suspended in distilled water with an OD_600_ = 0.2, respectively. Leaves of three biological replicates for each treatment were collected after 48 h and frozen in liquid nitrogen.

### 4.2. Generation of the Y2H Prey Library and Bait Constructs

Total RNA was isolated using phenol/chloroform extraction from leaves of ‘Hongyang’ and ‘Jinkui’ inoculated with Psa and water, respectively. Equal amount of RNA from the different treatments was mixed together. cDNA was synthesized using the SMART^TM^ cDNA Library Construction Kit (Clontech, Mountain View, CA, USA) and purified with the CHROMA SPIN-1000 column (Clontech, USA). The cDNA was then merged into the pGADT7 vector (Clontech, USA) according to the manufacturer’s protocol. The recombinant plasmid was transformed into *Escherichia coli*. After dilution (1:100), the *E. coli* cells were spread on LB media (Amp^+^) and incubated at 37 °C for 12 h. After incubation, 5000 colonies were calculated and the total colony forming units (cfu) was 1 × 10^7^. Twenty-four colonies were randomly amplified by PCR, and the insert size was mostly from 1.0 kb to 7.5 kb.

The complete coding sequence of PUB23 was amplified based on our previous transcriptome sequencing results, and inserted into pGBKT7 vector (Clontech, USA). To evaluate self-activation of PUB23, pGBKT7-PUB23 and pGBKT7 vectors were transformed into yeast strain AH109 separately, using the LiAc technique as described in the MATCHMAKER User Manual (Clontech, USA). The transformed yeast cells were plated on SD/-Trp medium and cultured at 30 °C for 3–5 days. Three colonies were randomly selected and copied onto SD/-Trp, SD/-Trp/-His, SD/-Trp/-His/-Ade and SD/-Trp/-His/-Ade+x-α-gal plates and cultured at 30 °C for 3–5 days.

### 4.3. Library Screening and Sequence Analysis

Plasmids of the prey and bait constructs were co-transformed into the AH109 strain according to the Protocol Handbook (Clontech, USA) and plated on SD/-Trp/-Leu/-His medium. After culture, positive clones were amplified by PCR and sequenced. The resulting sequences were analyzed using BLAST (https://blast.ncbi.nlm.nih.gov, accessed on 23 November 2023) with a fully automated procedure. Successfully sequenced clones were retransferred onto SD/-Trp/-Leu, SD/-Trp/-Leu/-His, SD/-Trp/-Leu/-His/-Ade and SD/-Trp/-Leu/-His/Ade+x-α-gal plates and cultured at 30 °C for 3–5 days. Clones successfully growing with blue color were recorded. Sequence of the positive clone WT-194 (GT1) was used as query sequence to perform BLAST searches against kiwifruit genome database (accession: GCA 003024255.1) and obtain the complete coding sequence. Protein sequences of GT factors in *Populus alba*, *Vitis vinifera*, *Theobroma cacao*, *Solanum tuberosum*, and *Arabidopsis thaliana* along with WT-194 were used to conduct multiple sequence alignment and generate a phylogenetic tree with neighbor-joining method [61] using MEGA 11 software package. Structure of GT1 was analyzed the SWISS-MODEL (https://swissmodel.expasy.org/, accessed on 23 November 2023).

### 4.4. In Vitro Pull-Down Assay

The coding sequence of PUB23 was cloned into vector pD2P (Clontech, USA) in frame with FLAG tag. Coding sequence (1089 bp) of the positive interacting protein (GT1) was cloned into pD2P in frame with 6×His tag. The recombined FLAG-PUB23 and His-GT1 were expressed by cell free expression system according to the protocol handbook (Nanjing Ruiyuan Biotech, Nanjing, China) and purified with FLAG-tag magnetic beads or His-tag magnetic beads, respectively. The purified FLAG-PUB23 and His-GT1 were co-incubated and precipitated with FLAG-tag magnetic beads or His-tag magnetic beads, respectively. Components were separated on a SDS-PAGE gel and immunoblotted with anti-FLAG (Abbkine, Wuhan, China; no. A02010) or anti-His (Abbkine, China; no. D-AKE2054) antibody.

### 4.5. In Vivo Co-Immunoprecipitation Assay

The coding sequence of PUB23 was cloned into vector pCAMBIA-3×FLAG and fragment of GT1 coding sequence was cloned into pBin-GFP2. The two recombinant vectors were transformed into Agrobacterium GV3101, respectively. Culture containing GFP-GT1 and FLAG-PUB23 and culture containing pBin-GFP2 and FLAG-PUB23 were injected into *Nicotiana benthamiana*, respectively. Total proteins were extracted and purified with GFP-tag magnetic beads or FLAG-tag beads. Components were analyzed by SDS-PAGE and immunoblotted with anti-GFP (Abbkine, China; no. D-AKE2024) or anti-FLAG antibody.

### 4.6. Virus-Induced Gene Silencing

Virus-induced gene silencing (VIGS) system based on the Tobacco Rattle Virus was used to silence *PUB23*. A 300 bp fragment of *PUB23* which shared no homology with any other sequence annotated in *A. chinensis* was amplified and cloned into pTRV2 vector. The recombinant pTRV2-PUB23, empty pTRV2, and pTRV1 were transformed into Agrobacterium GV3101, respectively, using the free-thaw method. Agrobacterium containing pTRV2-PUB23 and pTRV2 was mixed with Agrobacterium containing pTRV1 in a 1:1 ratio, respectively. The mixed cultures were used to inoculate leaves of kiwifruit ‘Hongyang’ and ‘Jinkui’ by pressure infiltration method. The materials were then injected with Psa in stem and held in in a growth room. Leaves of three biological replicates for each treatment were collected at 24, 48, and 96 h post Psa injection.

### 4.7. Quantitative RT-PCR and ROS Measurement

Total RNA was extracted from the leaves collected above. The first-strand cDNA was synthesized using the PrimeScript^TM^ RT reagent Kit with gDNA Eraser (Perfect Real Time) kit (TaKaRa, Dalian, China). Primers (Appendix A) for *GT1* were designed based on its sequence using Primer5 software. Primers for resistant genes *PR1*, *RIN4* and *ICS1* were as previously reported. The qRT-PCR reaction was performed using the SYBR^®^ *Premix Ex Taq*^TM^ (TaKaRa, China). Relative expression of the target genes was calculated using the 2^−ΔΔCt^ method [62] with three technical replicates. Hydrogen peroxide was determined using a commercial kit according to the manufacturer’s instruction (Genepioneer Biotechnologise, Hong Kong, China). The superoxide anion production rate was measured by a luminol-based assay as described by Wang and Luo [63]. Data were analyzed by one-way ANOVA.

## 5. Conclusions

PUB proteins have been reported to regulate plant immunity including PTI and ETI through interaction with other proteins. In this study, PUB23 was found to physically interact with GT1, a trihelix transcription factor, in vitro and in vivo. Silencing of *PUB23* led to the up-regulation of *GT1*. Immune responses of PAMP-triggered immunity (PTI), including a higher expression level of defense marker genes *PR1* and *RIN4* and increased accumulation of hydrogen peroxide and superoxide anion, were induced in PUB23-silenced plants. These findings show that the U-box type E3 ubiquitin ligase PUB23 in kiwifruit acts as a negative regulator of immune responses against Psa.

## Figures and Tables

**Figure 1 ijms-25-01930-f001:**
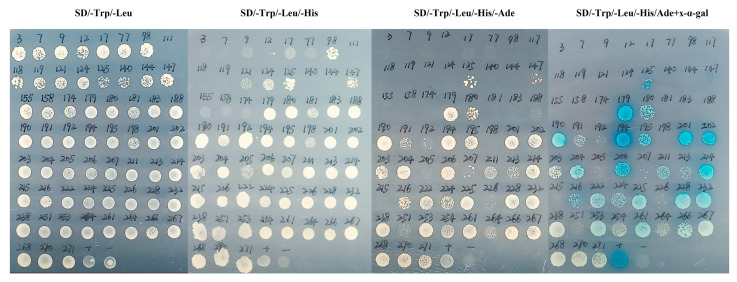
Y2H screening of potential proteins interacting with PUB23. By Y2H assay, 272 positive clones grew on the SD/-Trp/-Leu/-His medium, and 59 ones were successfully sequenced. To verify the interaction, all the 59 clones were plated onto selective mediums (SD/-Trp/-Leu, SD/-Trp/-Leu/-His, SD/-Trp/-Leu/-His/Ade and SD/-Trp/-Leu/-His/Ade+x-α-gal).

**Figure 2 ijms-25-01930-f002:**
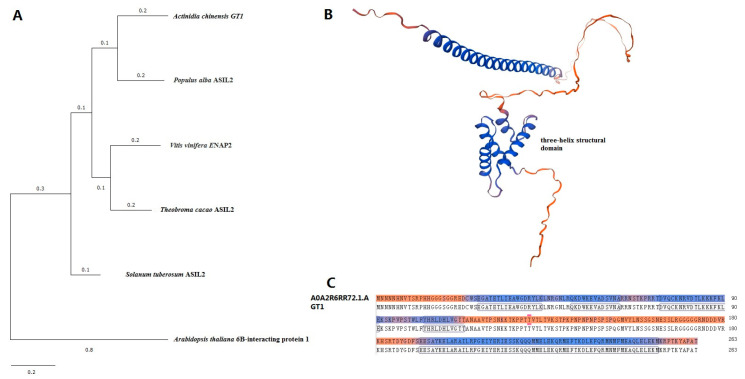
Sequence analysis of GT1. (**A**) Phylogenetic analysis of GT1 and homologs from other plants. The phylogenetic tree was generated by neighbor-joining method using MEGA11 program with 1000 boot strap trials. (**B**) Protein structure prediction of GT1. The structure was predicted by SWISS-MODEL. (**C**) Amino acid sequence alignment between GT1 and A0A2R6RR72.1.A.

**Figure 3 ijms-25-01930-f003:**
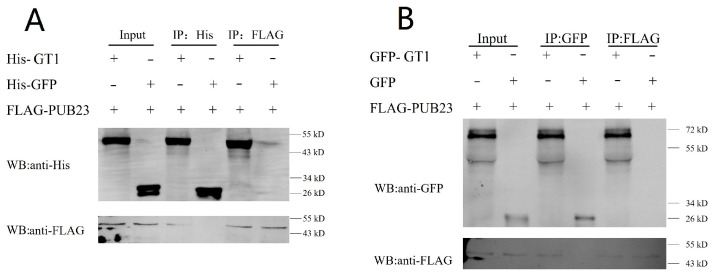
Kiwifruit PUB23 interacts with WT-GT1 in vitro and in vivo. (**A**) In vitro pull-down assay. FLAG-PUB23 and His-GT1 were expressed by cell free expression system, co-incubated, and precipitated with FLAG-tag magnetic beads or His-tag magnetic beads, respectively. The eluted proteins were then resolved by SDS-PAGE and probed with anti-FLAG or anti-His antibody. (**B**) In vivo co-immunoprecipitation assay. FLAG-PUB23 and GFP-GT1 were expressed in *N. benthamiana* and purified with GFP-tag magnetic beads or FLAG-tag beads. Components were immunoblotted with anti-GFP or anti-FLAG antibody.

**Figure 4 ijms-25-01930-f004:**
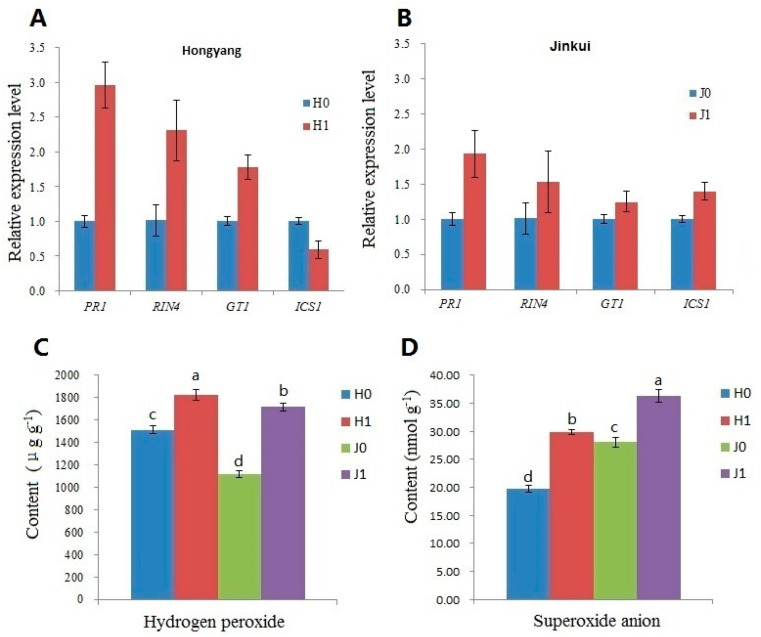
Influences of *PUB23* silencing on expression of defense marker genes and ROS accumulation. *PUB23* in ‘Hongyang’ and ‘Jinkui’ was silenced using the virus-induced gene silencing system. The materials were then injected with Psa in stem, and leaves were collected at 48 h post Psa injection. Data are shown as mean ± SD. Different letters indicate significantly different values *p* < 0.05. (**A**) Relative expression of *PR1*, *RIN4*, *GT1*, and *ICS1* in ‘Hongyang’ kiwifruit. (**B**) Relative expression of *PR1*, *RIN4*, *GT1*, and *ICS1* in ‘Jinkui’ kiwifruit. (**C**) Content of hydrogen peroxide. (**D**) Content of superoxide anion.

**Table 1 ijms-25-01930-t001:** Predicted products of positive clones.

Clone	Descriptions	E Value
WT-179	PREDICTED: *Camellia sinensis* 3-isopropylmalate dehydratase large subunit, chloroplastic (LOC114261229), mRNA	0
WT-180	None	
WT-190	PREDICTED: *Camellia sinensis* long-chain base biosynthesis protein 2a (LOC114285736), mRNA	0
WT-191	*Actinidia eriantha* clone FL10 thaumatin protein mRNA, complete cds	0
WT-192	PREDICTED: *Punica granatum* polyubiquitin (LOC116204478), mRNA	1 × 10^−131^
**WT-194**	Trihelix transcription factor [*Actinidia chinensis* var. chinensis]	3 × 10^−98^
WT-195	PREDICTED: *Camellia sinensis* DEAD-box ATP-dependent RNA helicase 35 (LOC114295640), transcript variant X11, mRNA	0
WT-201	PREDICTED: *Camellia sinensis* BEL1-like homeodomain protein 1 (LOC114279486), transcript variant X4, mRNA	7 × 10^−165^
WT-202	PREDICTED: *Camellia sinensis* coatomer subunit beta-1 (LOC114311215), mRNA	0
WT-204	PREDICTED: Quercus lobata carbamoyl-phosphate synthase small chain, chloroplastic (LOC115995429), mRNA	0
WT-206	PREDICTED: Ipomoea nil probable transcription factor At5g28040 (LOC109180351), mRNA	2 × 10^−29^
WT-211	PREDICTED: *Camellia sinensis* dnaJ protein homolog (LOC114258630), transcript variant X1, mRNA	0
WT-214	*Actinidia chinensis* FUL mRNA, complete cds	0
WT-228	PREDICTED: *Camellia sinensis* BEL1-like homeodomain protein 2 (LOC114301811), transcript variant X2, mRNA	5 × 10^−111^
WT-216	Disease resistance protein [*Actinidia chinensis* var. chinensis]	0
WT-232	PREDICTED: *Camellia sinensis* ribosomal RNA-processing protein 7 homolog A-like (LOC114264632), mRNA	1 × 10^−172^

## Data Availability

The data presented in this study are available within the article text and figures.

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
