# Peer review of "E3 Ubiquitin Ligase PUB23 in Kiwifruit Interacts with Trihelix Transcription Factor GT1 and Negatively Regulates Immune Responses against Pseudomonas syringae pv. actinidiae"

_ijms, 2024, doi:10.3390/ijms25031930_

Round 1

Reviewer 1 Report

Comments and Suggestions for Authors

Manuscript "E3 ubiquitin ligase PUB23 in kiwifruit interacts with trihelix transcription factor GT1 and negatively regulates immune responses against Pseudomonas syringae pv. actinidiae" is very interesting.

General comments:
Authors analyzed the role PUB23 played in immune responses of kiwifruit to Psa infection.
PUB23 was demonstrated to interact with a trihelix transcription factor GT1 which belongs to the GT factor family.

Detailed comments:
The introduction is written very well. The description of the material and method is very comprehensive. Unfortunately, the description of the statistical methods used was lacking.
Figure 2: What method was used to calculate similarity.
Figure 4: Statistical analysis of the presented results is missing. Homogeneous groups should be added.

Paper needs minor revision.

Author Response

1. Summary

Thank you very much for taking the time to review this manuscript. Please find the detailed responses below and the corresponding revisions/corrections in track changes in the re-submitted files.

2. Point-by-point response to Comments and Suggestions for Authors

Comments 1: Unfortunately, the description of the statistical methods used was lacking.

Response 1: We have provided more details in ‘materials and method’ part and marked the biological replicates.

Comments 2: Figure 2: What method was used to calculate similarity.

Response 2: We conducted multiple sequence alignment and generate a phylogenetic tree with neighbor-joining method [61] using MEGA 11 software.

Reference 61.       Saitou, N.; Nei, M. The neighbor-joining method: a new method for reconstructing phylogenetic trees. Mol. Biol. Evol. 1987, 4, 406–425.

Comments 3: Figure 4: Statistical analysis of the presented results is missing.

Response 3: Statistical analysis has been added.

Reviewer 2 Report

Comments and Suggestions for Authors

This manuscript by Wang et al. describes the interaction between an E3 ligase and a transcription factor in kiwifruit. E3 ligases are important for protein degradations.  The E3 ligase, PUB23, negatively regulates the expression of GT1. In addition, silencing PUB23 further leads to enhanced immune responses of PAMP-triggered immunity. These findings show the importance of PUB23 in kiwifruit. The following points need to be addressed:

1.     Missing references: Line 44-45, Line 62-63, Line 99-100 (cite the original publication of GT factors), and Line 113 (SWISS-MODEL).

2.     The authors did comprehensive analysis on GT1 in Figure 2. For example, they generated phylogenetic tree with MEGA11 and compared the amino acid sequence with another TF, A0A2R6RR72.1.A. What did you learn from the analysis? Can PUB23 interact with other trihelix TF?

3.     In Figure 3, the authors tried to show the interaction between PUB23 and GT1. GFP was used as a negative control. However, in Figure A, the inputs were GT1 with GFP, and PUB23 with GFP. In addition, the MW of GFP should be around 27 kDa. Is it anti-Flag instead of anti-GFP? Figure 3A need further elaboration and better explanation.

4.     Change the scale of y-axis of Figure 4B. Is the upregulation of GT1 significant in Jinkui? Any thoughts on why ICS1 had different patterns? The Figure legend needs to be improved.

5.     Add Figure 4 C,D in Section 2.5.

6.     Have you ever tested the protein level and ubiquitination level of GT1 by PUB23 silencing? Can PUB23 ubiquitinate GT1 in in-vitro ubiquitination assay?

7.     Provide more details in Materials and Methods. For example, in 4.4, how much protein did you use in the pulldown assay? What’s the catalog # of antibodies? How many biological replicates did you do for Quantitative RT-PCR?

8.     Provide more details in supplementary information. For example, the original western blots, raw data of each replicate.

Minor point:

1.     There are a few typos: Line 276: caner - canker, Line 330: in vitro – in vivo

2.     Check the reference format.

Comments on the Quality of English Language

Minor editing of English language required

Author Response

1. Summary

Thank you very much for taking the time to review this manuscript. Please find the detailed responses below and the corresponding revisions/corrections in track changes in the re-submitted files.

2. Point-by-point response to Comments and Suggestions for Authors

Comments 1: Missing references: Line 44-45, Line 62-63, Line 99-100 (cite the original publication of GT factors), and Line 113 (SWISS-MODEL).

Response 1: We checked all the references again and added the references.

Comments 2: The authors did comprehensive analysis on GT1 in Figure 2. For example, they generated phylogenetic tree with MEGA11 and compared the amino acid sequence with another TF, A0A2R6RR72.1.A. What did you learn from the analysis? Can PUB23 interact with other trihelix TF?

Response 2: We analyzed the structure of GT1 to confirm it does have a conserved trihelix domain like other GT factors. We constructed the phylogenetic tree to check if homologous GT factors participate in plant immune and found the most closed homolog ASIL2 in Solanum tuberosum was involved in embryo maturation.

Comments 3: In Figure 3, the authors tried to show the interaction between PUB23 and GT1. GFP was used as a negative control. However, in Figure A, the inputs were GT1 with GFP, and PUB23 with GFP. In addition, the MW of GFP should be around 27 kDa. Is it anti-Flag instead of anti-GFP? Figure 3A need further elaboration and better explanation.

Response 3: I am really sorry for my mistake in picture processing when labeling the ‘+ -‘ and I have replaced the figure with a new one. The inputs were GT1 with PUB23, and GFP with PUB23 as described in ‘materials and methods’. So the doubt about the MW of GFP can be explained. It is around 27 kDa in the new picture.

Comments 4: Change the scale of y-axis of Figure 4B. Is the upregulation of GT1 significant in Jinkui? Any thoughts on why ICS1 had different patterns? The Figure legend needs to be improved.

Response 4: The scale of y-axis of Figure 4B was changed as Figure 4A. The upregulation of GT1 in Jinkui is significant although with small amplitude. Actually we cannot explain this difference but we propose it may be related with the accumulation level of SA and it need farther research. The Figure legend has been improved.

Comments 5: Add Figure 4 C,D in Section 2.5.

Response 5: Figure 4 C,D has been added in Section 2.5.

Comments 6: Have you ever tested the protein level and ubiquitination level of GT1 by PUB23 silencing? Can PUB23 ubiquitinate GT1 in in-vitro ubiquitination assay?

Response 6: Unfortunately we did not do these works.

Comments 7: Provide more details in Materials and Methods. For example, in 4.4, how much protein did you use in the pulldown assay? What’s the catalog # of antibodies? How many biological replicates did you do for Quantitative RT-PCR?

Response 7: More details have been provided in Materials and Methods shown in the uploaded manuscript.

Comments 8: Provide more details in supplementary information. For example, the original western blots, raw data of each replicate.

Response 8: The original data was sent to the editors by email.

Minor point:

Comments 1: There are a few typos: Line 276: caner - canker, Line 330: in vitro – in vivo

Response 1: The typos have been corrected.

Comments 2: Check the reference format.

Response 2: The reference format has been checked again.

Round 2

Reviewer 2 Report

Comments and Suggestions for Authors

Thank you for addressing the comments. However, there are still a few things need to be addressed.

The anti-GFP western blot in Figure 3A showed bands between 43 and 55 kDa which were not corresponding to GFP's molecular weight. It's GFP-GT1. However, GT1 didn't have GFP tag in this pulldown experiment. Could you explain that? Also, what's the band around 50 kDa in anti-GFP antibody in Figure 3B?

Include the original data to the supplementary information.

Minor point:

Line 161: Further - Farther

Author Response

Thank you very much for taking the time to review this manuscript. I made a mistake when labeling the antibodies in WB in figure 3A. Anti-GFP should be anti-FLAG and I have corrected it. So bands between 43 and 55 kDa in figure 3A were PUB23 with FLAG tag as described in ‘methods and materials’. Band around 50 kDa in anti-GFP antibody in figure 3B was the heavy chains of primary antibody used in WB. This is a common phenomenon in WB.

Round 3

Reviewer 2 Report

Comments and Suggestions for Authors

Thank you for addressing my questions. However, I still have a few questions based on your explanation. I agreed that you could see heavy and light chains of the antibody in pulldown experiments. Why would the heavy chain only be observed in the first input lane (GFP-GT1 and FLAG-PUB23) but not the second one (GFP and FLAG-PUB23)? In addition, the input should be the total proteins from Nicotiana benthamiana. The input lysate should not have any antibody in it, right? GT1 can be pulled down by PUB23 (vice versa) based on the third and fifth lanes, though the protein level of PUB23 was much lower than GT1. The heavy chains were there too. However, there were not any heavy chains in either lane 4 or 6. Any explanation? Also, the heavy chain was not observed in the anti-FLAG antibody western at all. Since they were both anti-mouse antibodies, there should be heavy chains around 50 kDa there as well. 

Author Response

Thank you for discussion about the results. I discuss with my colleagues and think it is not heavy chains of antibody due to its smaller MW. Protein of this band showed the same feature as GFP-GT1. So we think it is a product of GFP-GT1 incomplete degradation or a protein which forms a complex with GFP-GT1 and can combine with antibody. We are also looking forward to your opinion.

Round 4

Reviewer 2 Report

Comments and Suggestions for Authors

Thanks for checking. However, there was no truncated His-GT1 in Figure 3A. Add more context in main text. Did you cross link your antibody to the beads? Is that why you didn't see any heavy and light chains? If you did, also state in method or SI.

Author Response

Thanks for  your opinion. The different expression environment may lead to a truncated GFP-GT1 but no truncated  His-GT1.

More context  was added in main text on this point. 

We did not cross link the antibody to the beads.